# Clinical Recommendations to Manage Gastrointestinal Adverse Events in Patients Treated with Glp-1 Receptor Agonists: A Multidisciplinary Expert Consensus

**DOI:** 10.3390/jcm12010145

**Published:** 2022-12-24

**Authors:** Juan J. Gorgojo-Martínez, Pedro Mezquita-Raya, Juana Carretero-Gómez, Almudena Castro, Ana Cebrián-Cuenca, Alejandra de Torres-Sánchez, María Dolores García-de-Lucas, Julio Núñez, Juan Carlos Obaya, María José Soler, José Luis Górriz, Miguel Ángel Rubio-Herrera

**Affiliations:** 1Department of Endocrinology and Nutrition, Hospital Universitario Fundación Alcorcón, 28922 Madrid, Spain; 2Department of Endocrinology and Nutrition, Hospital Universitario Torrecárdenas, 04009 Almería, Spain; 3Department of Internal Medicine, University Hospital of Badajoz, 06080 Badajoz, Spain; 4Department of Cardiology, University Hospital la Paz, IdiPAZ, Biomedical Research Center-Cardiovascular Diseases (CIBERCV-ISCIII), 28046 Madrid, Spain; 5Health Centre Casco Antiguo Cartagena, Primary Care Research Group, Biomedical Research Institute of Murcia (IMIB), 30201 Cartagena, Spain; 6Department of Internal Medicine, Costa del Sol Hospital, 29603 Marbella, Spain; 7Department of Cardiology, Valencia Clinic University Hospital, Instituto de Investigación Sanitaria (INCLIVA), 46010 Valencia, Spain; 8Health Centre la Chopera, 28100 Madrid, Spain; 9Nephrology and Kidney Transplantation Research Group, Nephrology Department, Vall d’Hebron Institut de Recerca (VHIR), Vall d’Hebron Hospital Universitari, 08035 Barcelona, Spain; 10Nephrology Department, Valencia Clinic University Hospital, Instituto de Investigación Sanitaria (INCLIVA), Universitat de València, 46010 Valencia, Spain; 11Department of Endocrinology and Nutrition, San Carlos Clinical Hospital, Health Research Institute of the San Carlos Clinical Hospital (IDISSC), 28040 Madrid, Spain

**Keywords:** type 2 diabetes, obesity, glucagon-like peptide-1 receptor agonists, gastrointestinal adverse events, guidelines

## Abstract

Glucagon-like peptide-1 receptor agonists (GLP-1 RAs) are indicated in type 2 diabetes and obesity for their high efficacy in controlling glycaemia and inducing body weight loss, respectively. Patients may develop gastrointestinal adverse events (GI AEs), namely nausea, vomiting, diarrhoea and/or constipation. To minimize their severity and duration, healthcare providers (HCPs) and patients must be aware of appropriate measures to follow while undergoing treatment. An expert panel comprising endocrinologists, nephrologists, primary care physicians, cardiologists, internists and diabetes nurse educators convened across virtual meetings to reach a consensus regarding these compelling recommendations. Firstly, specific guidelines are provided about how to reach the maintenance dose and how to proceed if GI AEs develop during dose-escalation. Secondly, specific directions are set about how to avoid/minimize nausea, vomiting, diarrhoea and constipation symptoms. Clinical scenarios representing common situations in daily practice, and infographics useful to guide both HCPs and patients, are included. These recommendations may prevent people with T2D and/or obesity from withdrawing from GLP-1 RAs treatment, thus benefitting from their superior effect on glycaemic control and weight loss.

## 1. Introduction

Glucagon-like peptide-1 receptor agonists (GLP-1 RAs) have represented a paradigm shift in the treatment of type 2 diabetes (T2D) and obesity. The incretin effect induced by GLP-1 RAs allows for glycaemic control in a glucose-dependent manner more efficiently than other therapeutic classes without increasing the risk of hypoglycaemia [1]. Interestingly, some of them are able to cross the blood–brain barrier and act on the brain to stimulate satiety [2], which leads to food intake reduction and, consequently, body weight loss, which occurs at the expense of fat mass. As a result, the risk of progression to T2D decreases, and improvements in lipid profile, blood pressure or sleep apnoea have been observed [3]. GLP-1 RAs also exert pleiotropic actions with metabolic, hepatic, renal and cardiovascular beneficial effects [4,5,6,7]. Of note, a recent meta-analysis encompassing those clinical trials focused on the cardiovascular safety of GLP-1 RAs found significant reductions in MACE3 (a composite of cardiovascular death, non-fatal myocardial infarction, and non-fatal stroke), hospitalization for heart failure and progression of chronic kidney disease [8]. The currently commercialized GLP-1 RAs with indications for T2D or obesity are summarized in Appendix A. 

Ample clinical evidence arising from trials and real-world scenarios highlights that the most frequent adverse events (AEs) associated with GLP-1 RA are those of a gastrointestinal (GI) nature, namely nausea, vomiting, diarrhoea and constipation. According to the literature addressing clinical trials, GI AEs usually develop in 40–70% of treated patients, although they have sometimes been reported in up to 85% [9,10,11,12,13,14,15,16,17]. 

GI AEs arise irrespective of the half-life (long/short action) or route of administration (subcutaneous/oral) of the chosen GLP-1 RA. They are usually transient, typically starting during the dose-escalation period and generally resolving shortly after the maintenance dose is reached, and, in most cases, they are mild to moderate in severity. A recent report summarizing the results of several trials reported that the majority (99.5%) of documented GI AEs in people with obesity on GLP-1 RA treatment were non-serious [18]. A meaningful body of evidence in real-world settings roughly replicates these observations [19,20,21,22]. Nevertheless, it is essential that patients and healthcare professionals (HCPs) are aware of the right procedures to follow to prevent GI AEs from arising or, if they occur, to mitigate their effects and improve adherence and persistence to the treatment. 

GI AEs may lead to the temporary or permanent discontinuation of GLP-1 RA treatment. Although interruption has been reported to occur in up to 12% of GLP-1 RA-treated patients (vs. ~2% in those treated with placebo) [9,10,11,12,13,14,15,16,17,18], permanent discontinuations range between 1.6–6% of treated patients (vs. <1% with placebo), according to clinical trial programs [9,15,18]. In the real-world setting, persistence with GLP-1 RA therapy has been found to range between 40% and 60% and between 34% and 67% at 180 and 360 days, respectively [23,24,25]. The results seem to be better with once-weekly administered GLP-1 RAs compared to those requiring once-daily injections [26,27,28]. The adequate management of GI AEs would potentially improve a patient’s experience while undergoing treatment with GLP-1 RA, preventing discontinuations secondary to GI AEs.

The literature addressing the management of GI AEs in clinical practice associated with the use of GLP-1 RA from a multidisciplinary perspective is scarce [29]. For this reason, we formed a multidisciplinary team consisting of HCPs of several specialties and a diabetes nurse educator (DNE) with experience in the management of patients treated with GLP-1 RA in their daily practice. The main goal of this document is to reach a consensus regarding patient education and procedures to minimize the frequency and intensity of GLP-1 RA-associated GI AEs. Hopefully, this multidisciplinary consensus may contribute to expanding our knowledge of the practical use of GLP1-RAs, providing useful tools for physicians to use when managing people with T2D or overweight/obesity with this therapeutic class. More importantly, the better management of GI AEs could potentially have a positive impact on the adherence/persistence, effectiveness of treatment and quality of life (QoL) among GLP-1 RA-treated patients. 

## 2. Methods

A multidisciplinary expert panel comprising three endocrinologists, two nephrologists, two primary care physicians (PCPs), two cardiologists, two internists and one DNE, all of them with clinical expertise in the management of people with T2D or obesity with GLP-1 RA-based therapies, convened across virtual meetings to reach a consensus. The participants performed a comprehensive literature search using prespecified keys (Appendix A). The experts agreed that GI AEs should be the main topic to focus on when addressing the optimal medical care of patients treated with GLP-1 RA. The proper management of such occurrences is of paramount importance since it contributes significantly to treatment adherence and persistence and may improve QoL. Hepatobiliopancreatic AEs are also discussed in brief because, although they are extremely infrequent, it is important to exercise caution with those patients with a history of pancreatic and gallbladder disorders. Finally, widely accepted myths and erroneous considerations concerning other AEs and profiles of subjects wrongly considered unfit to receive this medication are also reviewed to ensure that as many patients as possible can benefit from treatment with GLP-1 RA. 

## 3. AEs Most Frequently Associated with GLP-1 RAs 

As anticipated, the AEs that were most frequently associated with GLP-1 RAs are those of a GI nature. Table 1 shows a summary of their frequency of occurrence in phase III clinical trials. Among the most common GI side effects, namely nausea, vomiting, diarrhoea and constipation, nausea appears systematically as the most frequent event in all clinical trials. The prevalence of the other GI AEs is lower. Overall, the onset of GI AEs is slightly higher in those trials designed to assess the efficacy and safety of GLP-1 RAs in people with obesity, which may be ascribed to the fact that doses are higher than those used in clinical trials in people with T2D. It is worth mentioning that long-acting agents have been associated with less nausea and vomiting but with more diarrhoea [30], which might be explained by a more sustained effect of these compounds on GLP-1 intestinal receptors [31]. Finally, it is worth mentioning that flatulence may occasionally appear, although studies reporting its frequency are lacking.

## 4. Practical Guide to Follow When Initiating Treatment with GLP-1 RAs 

### 4.1. Patient Education Prior to GLP-1 RA Start 

Patient education in terms of how to take and deal with satiety once GLP-1 RAs are started is crucial for ensuring treatment compliance. It is accepted that persistence improves when weight is adequately managed and safe, straightforward treatments are used [32,33,34]. GLP-1 RA-based treatments comply with these characteristics. The proper education of patients by HCPs on their expectations when initiating treatment, how to prevent AEs, and how to treat them if they appear is particularly important. Patients have to learn that, although GI AEs may occur, these will be transient and of mild/moderate severity in the majority of cases and that following specific dietary recommendations will relieve symptoms (Table 2, Figure 1, Figure 2 and Figure 3). 

### 4.2. Overall Procedures

Compliance with the guidelines contained in Table 2 will be useful in avoiding the onset of GI AEs and mitigating their intensity in those cases where they develop. Indeed, it is important to *start low and go slow*: the lowest dose of medication should be used to start, and HCPs should ensure that patients follow their instructions closely. In those rare cases of particularly intense and/or persistent GI AEs, HCPs can apply further measures (Figure 4): If GI AEs appear during the dose-escalating phase, HCPs could modify the planned schedule by implementing one or several of the following points [35,36]:▪Extend the dose escalation phase duration (2–4 more weeks with the previous dose or temporary suspension).▪Avoid dose escalation while GI AEs persist.▪If a GI AE is experienced when moving up to a higher dose, go back on the lower one and stay on it for a few days. Then, increase the dose gradually, taking advantage of the multidose pen-injector when available.▪In the case of persistent tolerability limitations set a dose lower than the maximum one recommended by the technical data sheet as a maintenance dose. ▪Withhold treatment temporarily until the resolution of AEs and then resume treatment.


Appropriate management during the dose-escalation phase could help patients to develop tachyphylaxis, i.e., to avoid excessive delay in gastric emptying upon treatment with GLP-1 RAs, especially with the long-acting ones [37,38]. 

Start a differential diagnosis procedure to rule out underlying conditions causing the symptoms or their exacerbation [35].Make sure that the patient understands and complies with the guidelines regarding diet habits (Table 2, Figure 1 and Figure 2).Start symptomatic treatment focused on the specific GI AE (see below).Switching to another GLP-1 RA may be considered. Although reported data from trials and real-world series do not show definitive differences between GLP-1 RAs in terms of tolerability (Table 1), there are studies claiming that the tolerability profile may vary between different compounds [1,33,39]. In fact, the switching strategy has already been proposed in the context of the treatment of people with T2D [40,41]. It is advisable to start the new GLP-1 RA at its lowest escalation dose. If the patient is on treatment with semaglutide, switching the route of administration, i.e., from s.c. to oral semaglutide or vice versa, may be an option [42].

### 4.3. Specific Procedures

#### 4.3.1. Nausea 

Real-world studies confirm that nausea is the most frequent GI AE that arises upon initiation of GLP-1 RA [20,43,44]. The frequency varies among clinical trials, although it usually ranges between 15 and 50% (Table 1). Prevalence is higher during the first 4–5 weeks of treatment, when gastric emptying is more significantly delayed [45], decreasing thereafter [9,10,11,12,13,14,15,16,17]. Symptoms are usually moderate and disappear 8 days or less after their onset [18,46]. To alleviate symptomatology, the following measures are advisable [29]:○Comply with recommendations regarding intake habits and food composition (Table 2, Figure 1, Figure 2 and Figure 3).○If symptoms still persist, consider anti-emetic and/or prokinetic medications (Figure 4). Domperidone (10–20 mg three to four times daily, oral dosage, not in children < 12 years) should be used rather than metoclopramide, especially in older patients, to minimize the risk of extrapyramidal side effects [47]. Among substituted benzamides, cinitapride may be an alternative to metoclopramide [48]. In the event that oral semaglutide is being used, a period of 30 min must elapse between the administration of both medications. If drugs to mitigate nausea (or other GI AEs) are needed for over a month when the maintenance GLP-1 RA dose has been reached, a dose reduction should be considered for the patient to tolerate the drug with no need for pharmacological support [35].

#### 4.3.2. Vomiting 

Vomiting occurs at a frequency that usually ranges between 5 and 20%, i.e., lower than that reported for nausea (Table 1). Symptoms have been reported to disappear in 1–8 days [18,46]. Initial measures do not substantially differ from those already reviewed (Figure 1). In case of abnormally long persistence or unexpected severity, additional actions may be undertaken (Table 2, Figure 2, Figure 3 and Figure 4) [35]: ○Maintaining hydration is particularly important. ○Small amounts of food should be taken in more frequent meals.○Consider anti-emetic and/or prokinetic medications. Domperidone should be used rather than metoclopramide, as explained above [47]. ○In case of persistence and/or remarkable severity, and where the patient presents with dizziness, confusion and fatigue, standard procedures to clinically manage severe vomiting can be initiated. Although rarely, intravenous rehydration may be necessary. 

If the above measures are not as efficient as expected, either a decrease in GLP-1 RA dose or a temporary interruption of the treatment may be considered [35]. 

#### 4.3.3. Diarrhoea 

The frequency of diarrhoea also varies among trials, usually ranging between 5 and 25% (Table 1). Diarrhoea initiates during the first four weeks of treatment, after which the incidence notably decreases [15,49]. Symptoms have been reported to last for about three days in people with obesity treated with GLP-1 RA [18]. The patients must be encouraged to follow the specific recommendations to avoid/mitigate diarrhoea (Table 2, Figure 1, Figure 2 and Figure 3), which should already have been taught to them as part of their education prior to GLP-1 RA initiation. In the event of persistence, in spite of compliance with these and the other guidelines, probiotic and/or antidiarrheal supplements such as loperamide could be considered [29]. GLP-1 RA may exacerbate diarrhoea in patients receiving metformin treatment, especially if they are also taking omeprazole. In such cases, a reduction in the dose of metformin may be advisable (Figure 4). 

#### 4.3.4. Constipation 

Constipation generally occurs at frequencies in the range of 4–12%, i.e., lower than those documented with other GI AEs. Nevertheless, some trials have reported a prevalence of up to 25 and 35% in people with obesity (Table 1). Accordingly, real-world series confirmed that constipation occurs more frequently in patients with overweight/obesity than with T2D [50]. The onset can be in the first 16 weeks of treatment, particularly during the first 28 days [49,51,52]. Constipation symptoms have been reported to last longer than those of the other GI AEs. The symptoms persisted for a median duration of 47 days in people with obesity on GLP-1 RA therapy [18], which may be ascribed to the more chronic nature of the condition [35]. Secondary to the feeling of gastric fullness after receiving the drug, patients tend to reduce water intake, which may predispose them to this AE. Patients should be advised to increase their mobility, and intake of water and fibre, as well as to consider the use of stool softeners (Figure 1, Figure 2 and Figure 3). Clinicians should not discard a reduction in the GLP-1 RA dose upon a worsening of symptoms (Figure 4) [35].

## 5. Uncommon AEs 

Pancreatobiliary complications have been documented, albeit seldom. A comprehensive review encompassing 30 trials focusing on GLP-1 RA safety in people with T2D concluded that the risk of gallbladder complications or acute pancreatitis (AP) associated with this medication was generally low [53]. In people with obesity, the incidence of gallbladder-related events was always <3% [10,54]. Cholelithiasis was reported in 0–<1% patients in the majority of cohorts, regardless of having T2D or obesity [9,10,11,12,13,14,15,16,17]. Its association with GLP-1 RA use, although unusual, has been linked to a combination of factors. Among these, the relevant weight loss often experienced by people with obesity may promote biliary lithogenicity, such as after bariatric surgery. Other explanations might be a direct action of the drug on biliary secretion and/or the modification of gallbladder motility [55,56,57,58]. On the other hand, cholecystitis episodes were anecdotal [9,10,11,12,13,14,15,16,17]. Finally, although higher circulating levels of lipase and amylase were reported in patients on GLP-1 RA therapy in many trials, the increases were rarely higher than three or five-fold the upper limit of normal (ULN), respectively [9,10,11,12,13,14,15,16,17], returned to normal levels after medication withdrawn, and were poor predictors of AP [54].

A meta-analysis that included studies lasting for ≥24 months, with more than 9000 patients treated with GLP-1 RA during such period, did not find an association between GLP-1 RA therapy and AP either [59]. Another one covering 55 randomized controlled trials and five observational studies with more than 300,000 participants, drew the same conclusion [60]. Nevertheless, a recent meta-analysis encompassing up to 76 randomized controlled trials concluded that GLP-1 RA treatment was associated with a significant, albeit low, increased risk of gallbladder or biliary diseases (relative risk (RR), 1.37; 95% confidence interval (CI), 1.23–1.52). When the trials were stratified according to pathology, RR was 2.29 (95% CI, 1.64–3.18) in trials focused on people with obesity, near two-fold higher than that observed in trials concerning the treatment of T2D or other diseases (1.27; 95% CI, 1.14–1.43). Higher GLP-1 RA doses and, especially, the duration of treatment influenced the risk [61]. 

The proportion of patients who developed pancreatobiliary complications such as AEs in the course of the more significant clinical trials is summarized in Table 3. AP was always experienced by less than 1% of patients treated with GLP-1 RAs. Furthermore, many of the cases were reported in subjects with a previous history of pancreatitis or gallbladder disease. Thus, caution must be exercised in patients with these antecedents. Ursodeoxycholic acid may be administered to patients with a history of cholelithiasis. 

HCPs must be aware of the possibility of these rare side effects to act early and avoid the complications related to volume depletion, such as severe renal failure [62,63]. Patient education to recognize pancreatitis symptoms is also a recommended practice [29].

If developed, pancreatobiliary complications should be managed according to the guidelines of each centre. 

## 6. Myths or Reality? 

When GLP-1 RAs started to be used, the first observations of certain occurrences led to misconceptions and erroneous conclusions regarding either the risks associated with the use of these drugs or the profiles of patients who would not benefit from these therapies. 

### 6.1. Impact of GI AEs on Weight Loss

As previously stated, the most common GIs AEs included nausea, vomiting and diarrhoea. It has, therefore, been discussed whether the effects of GLP-1 RAs on weight loss are linked to these GI AEs. A study performed to assess whether or not there was a direct association between GI AEs and the extent of weight loss concluded that weight loss was largely independent of GI AEs [47,64].

### 6.2. Patient Profiles Falsely Considered Unfit

#### 6.2.1. Patients with Eating Disorders 

Many of these patients can benefit from GLP-1 RA therapy even though their dietary habits are poor. For example, patients with binge eating disorder or night-eating syndrome may benefit from GLP-1 RA therapy as long as they also receive proper psychological/psychiatric care. The exception are those patients with anorexia nervosa. GLP-1 RAs would also be contraindicated in patients with bulimia and self-induced vomiting until the psychiatric disorder was resolved using psychotherapy or pharmacologic means [65,66,67]. Indeed, HCPs should take care to provide those patients with eating disorders who are going to start GLP-1 RA treatment with educational tips aimed at promoting a healthy lifestyle. 

#### 6.2.2. Patients aged 75 Years Old or Older

The reduced cardiovascular and renal morbimortality associated with the use of GLP-1 RAs is beneficial for this particularly risky population [68,69,70,71,72,73]. Caution when using GLP-1 RAs in older patients, especially those ≥75 years old, has been suggested to prevent the risk of sarcopenia. Although, as said above, weight loss induced by GLP-1 RAs comes mainly from reductions in fat mass rather than lean mass, nutritional advice and exercise programmes to achieve muscle strengthening are recommended to minimize sarcopenic risk [74]. 

On the other hand, regarding the concern that GLP-1 RA doses indicated by the technical data sheet may be harmful to the elderly, it must be remarked that pivotal trials have repeatedly shown that the beneficial effects on weight loss and glycemic control can be achieved with no need to reach the recommended maintenance doses [9,10,11,12,13,14,15,16,17]. An intermediate dose may be considered as the maintenance dose in fragile patients, thus minimizing risks associated with GI AEs. The Clinical Frailty Scale may sometimes be useful to guide dose adjustment [75]. Finally, it must be remarked that GLP-1 RAs may be particularly beneficial for older patients receiving insulin treatment. The improved glycaemic control achieved with the new medication will probably make it possible to reduce insulin doses, which could decrease the risk of hypoglycaemic events and the consequences associated with these. 

#### 6.2.3. Patients with Upper Gastrointestinal Disease

A recently reported study focusing on patients with upper gastrointestinal disease, either chronic gastritis, gastroesophageal reflux disease, or both, undergoing GLP-1 RA treatment showed that this medication was well tolerated, with no need for dose adjustment. All of the AEs were mild-to-moderate, with no withdrawals because of them [76]. Patients with chronic intestinal pathology are candidates for GLP-1 RA use. Furthermore, preliminary data suggest that these compounds might play a beneficial role in intestinal diseases [77]. Nevertheless, the data sheets of several GLP-1 RAs do not recommend their administration to patients with severe gastrointestinal disease due to the limited experience in managing this condition. 

## 7. Clinical Scenarios of Interest

To address further practical considerations, four clinical scenarios are described that are aimed at providing clues for the better management of GI AEs in those patients at higher risk of these complications that are going to undergo GLP-1 RA-based therapy (Table 4). 

## 8. Conclusions

According to this multidisciplinary consensus, GLP-1 RAs can be considered easy-to-manage, highly effective medications with a good profile of tolerability to treat people with obesity or T2D. Patients should be educated to be aware that they may experience GI AEs, and that these will probably be mild to moderate in intensity and transient. Patients must also be educated on following a series of guidelines that will help them to prevent or, at least, mitigate GI AEs. HCPs must be aware that comprehensive dietary education, flexibility during the dose-escalation phase and appropriate symptomatic treatment of persistent GI AEs are factors of paramount importance to minimize the GI side effects associated with GLP-1 RA use. 

## Figures and Tables

**Figure 1 jcm-12-00145-f001:**
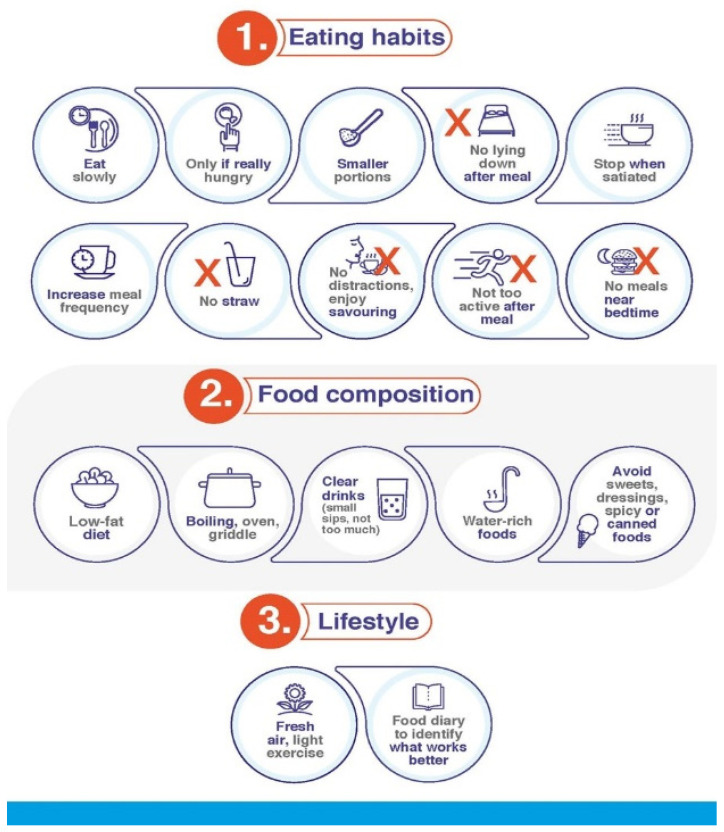
Minimizing occurrence/severity of GI AEs: patients general guidelines.

**Figure 2 jcm-12-00145-f002:**
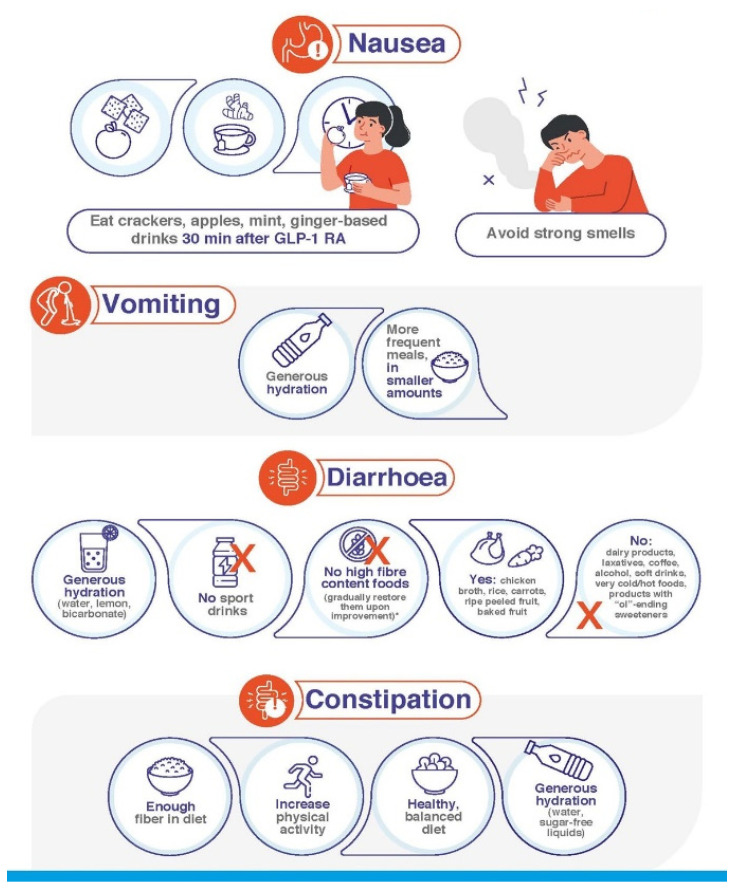
Additional specific guidelines for each separate GI AE.

**Figure 3 jcm-12-00145-f003:**
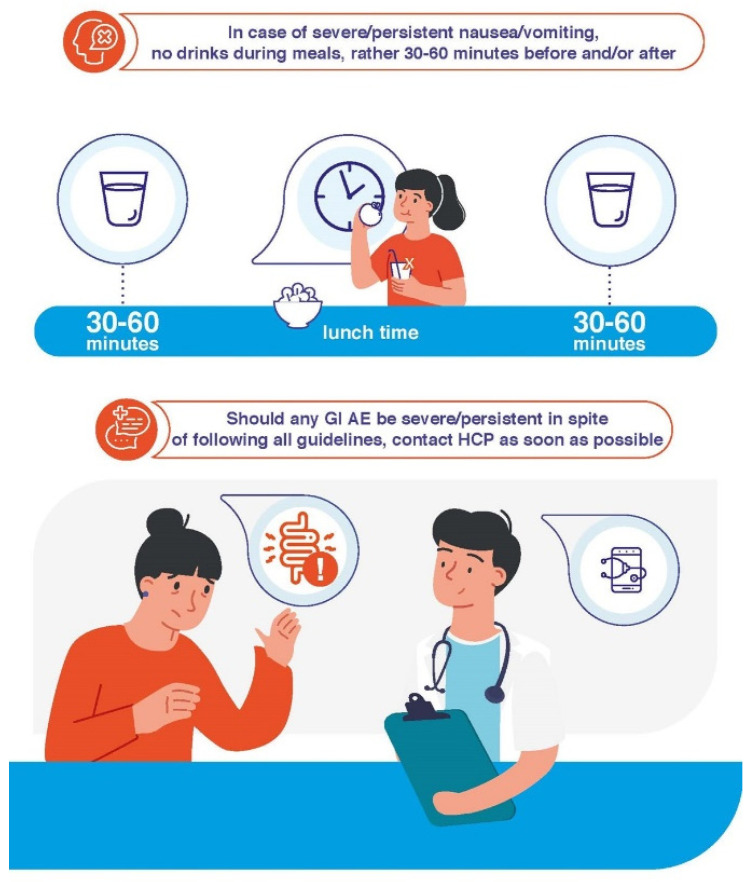
Additional specific guidelines for unusually severe/persistent GI AEs.

**Figure 4 jcm-12-00145-f004:**
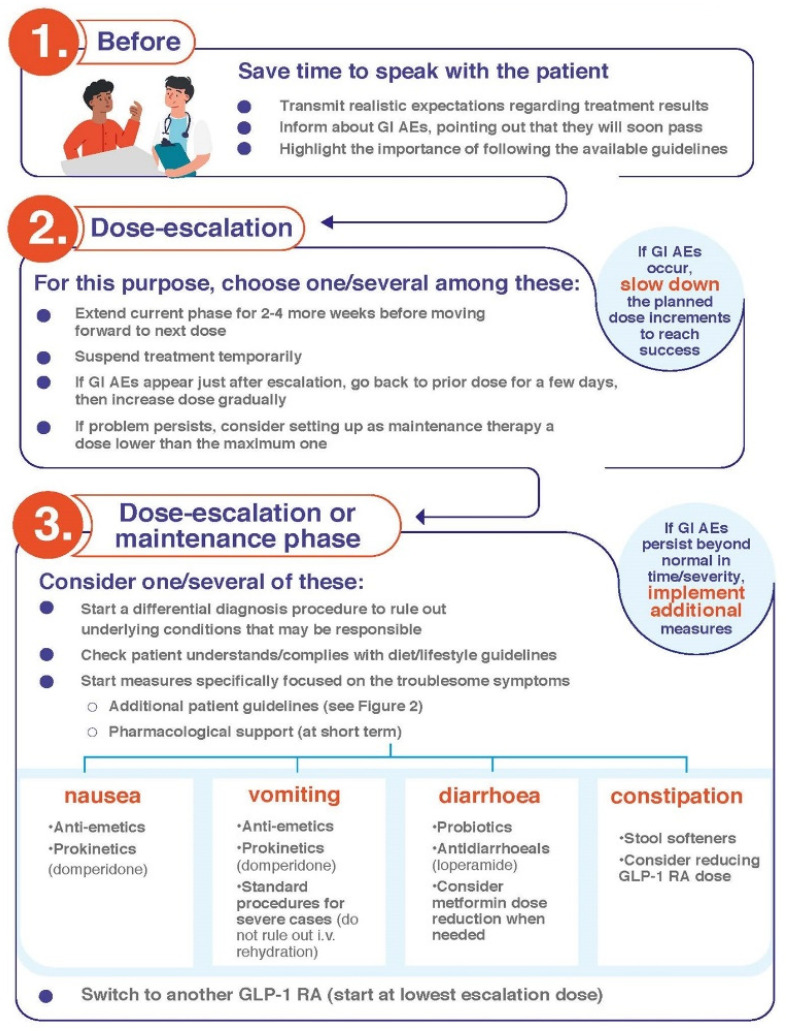
Minimizing occurrence/severity of GI AEs: the role of HCPs.

**Table 1 jcm-12-00145-t001:** Frequency of GI AEs in clinical trials with GLP-1 RA in people with obesity or T2D.

GLP-1 RA	Program	Refs	Patient Profile	Dose	Method of Administration	Nausea	Vomiting	Diarrhoea	Constipation
Semaglutide	SUSTAIN	9	T2D	1 mg	s.c. once weekly	15–24	7–15	7–19	4–7
Semaglutide	STEP	10	Obesity *	2.4 mg	s.c. once weekly	14–58	22–27	10–36	12–37
Semaglutide	PIONEER	11	T2D	14 mg	p.o. SID	8–23	6–12	5–15	7–12
Liraglutide	LEAD	12	T2D	1.8 mg	s.c. SID	10–40	4–17	8–19	11
Liraglutide	SCALE	13	Obesity *	3 mg	s.c. SID	27–48	7–23	16–26	12–30
Dulaglutide	AWARD	14	T2D	1.5 mg	s.c. once weekly	15–29	7–17	11–17	n.r.
Exenatide	DURATION	15	T2D	2 mg	s.c. once weekly	5–14	<1–6	5–11	1–8
Exenatide	—	16	T2D	10 µg	s.c. BID	35–59	9–14	4–9	5
Lixisenatide	GETGOAL	17	T2D	20 µg	s.c. SID	16–40	7–18	4–12	5†

Results are expressed as percentages of patients from the treatment cohort who experienced the AE at least once during the period of the study. Values correspond to the minimum and maximum values reported when considering all the studies of the program. * One out of 6 studies recruited people with obesity and T2D as well. † Data reported in one study only. BID, twice a day; GI AEs, gastrointestinal adverse events; GLP-1 RA, GLP-1 receptor agonist; n.r., not reported; p.o., oral; Refs, references; s.c., subcutaneous; SID, once a day; T2D, diabetes mellitus type 2.

**Table 2 jcm-12-00145-t002:** Guidelines for patients.

Recommendations to Minimize Occurrence/Severity of GI AEs when Starting GLP-1 RA Therapy
General recommendations
Observe the guidelines of the data sheet regarding posology and method of administration
* * Improve eating habits
Eat slowly
Eat only if you are really hungry
Eat smaller portions
Avoid lying down after having a meal
Stop eating in case of feeling of fullness
Increase meal frequency
Avoid drinking using a straw
Eat without distractions and enjoy savouring the food
Try not to be too active after eating
Avoid eating too close to bedtime
Adapt food composition to your requirements
Choose easy-to-digest food, low fat diets (focus on bland foods)
Use oven, cooking griddle or boiling
Increase fluid intake, especially clear, fresh drinks (in small sips), but no so much as to make you feel too full
Healthy food that contain water (soups, liquid yogurt, gelatin, and others)
Avoid sweet meals
Avoid dressings, spicy foods, canned food, sauces that are not home-cooked
Get some fresh air and do some light exercise
Keep a food diary, as it may be useful to identify foods or meal timings that make it worse
Additional recommendations for patients with nausea
Provided that 30 min have passed since the last GLP-1 RA dose, eat foods able to ease the symptoms of nausea, such as crackers, apples, mint, ginger root or ginger-based drinks
Avoid strong smells
Additional recommendations for patients with vomiting
Be particularly careful with hydration
Eat smaller amounts of food in more frequent meals
Additional recommendations for patients with diarrhoea
Generous hydration, for example with water, lemon and a teaspoon of bicarbonate
Avoid isotonic drinks intended to be used in the context of sport activities
Avoid dairy products, laxative juices or meals, coffee, alcoholic drinks, soft drinks, very cold or very hot foods, products with sweeteners ending in “ol” (sorbitol, mannitol, xylitol, maltitol), including candy and gum
Avoid (or temporarily reduce your intake of) foods with high fibre content * such as grain and seed products, such as grain cereals, nuts, seeds, rice, barley, whole grain bread or baked goods vegetables such as artichokes, asparagus, beans, cabbage, cauliflower, garlic and garlic salts, lentils, mushrooms, onions, sugar snap, snow peas skinned fruits, apples, apricots, blackberries, cherries, mango, nectarines, pears, plums
Eat chicken broth, rice, carrots, very ripe fruit without skin
Additional recommendations for patients with constipation
Ensure the amount of fibre in your diet is adequate
Increase physical activity
Ensure your diet is healthy and balanced
Drink generous amounts of water (or other sugar-free liquids)
Additional recommendations when GLP-1 RA are unusually severe or/and persistent
In case of persistence of nausea and/or vomiting, avoid drinks during meals, rather have them between 30 and 60 min before and/or after meals
If nausea, vomiting, diarrhoea and/or constipation persist in spite of following all the guidelines depicted above, inform HCP as soon as possible

* Gradually increase the amount of fibre intake once the symptoms improve. HCP, healthcare provider.

**Table 3 jcm-12-00145-t003:** Frequency of pancreatobiliary complications in clinical trials with GLP-1 RA in people with obesity or T2D.

GLP-1 RA	Program	Refs	Target Patient	Dose	Administration	Cholelithiasis	AP
Semaglutide	SUSTAIN	9	T2D	1 mg	s.c. once weekly	0–2	0-<1
Semaglutide	STEP	10	Obesity *	2.4 mg	s.c. once weekly	<1–3	0-<1
Semaglutide	PIONEER	11	T2D	14 mg	p.o. QD	0-<1	0-<1
Liraglutide	LEAD	12	T2D	1.8 mg	s.c. QD	0	0-<1
Liraglutide	SCALE	13	Obesity *	3 mg	s.c. QD	<1–1	0-<1
Dulaglutide	AWARD	14	T2D	1.5 mg	s.c. once weekly	0	<1
Exenatide	DURATION	15	T2D	2 mg	s.c. once weekly	0-<1	0-<1
Exenatide	—	16	T2D	10 µg	s.c. BID	0	0
Lixisenatide	GETGOAL	17	T2D	20 µg	s.c. QD	0	0

Results are expressed as percentages of patients from the treatment cohort who had the complication at least once during the period of the study. Values correspond to the minimum and maximum values reported when considering all the studies of the program. * One out of 6 studies recruited people with obesity and T2D as well. AP, acute pancreatitis; BID, twice a day; GI AEs, gastrointestinal adverse events; GLP-1 RA, GLP-1 receptor agonist; n.r., not reported; p.o., oral; Refs, references; s.c., subcutaneous; QD, once a day; T2D, diabetes mellitus type 2.

**Table 4 jcm-12-00145-t004:** Clinical scenarios of interest to exemplify GI AE management during GLP-1 RA use.

Scenario 1. Persistent Nausea in Middle-Aged Patients
Female patient, 52 y.o., T2D, 6 years’ duration on metformin treatment. The patient starts oral semaglutide 3 mg OD without having received guidelines to prevent GI AEs. Four weeks later, the dose is increased to 7 mg OD, and 4 weeks later, to 14 mg OD. Since then, occasional intense episodes of nausea associated with eating large meals that include fatty foods occur. The dose is maintained at 14 mg OD for 2 months, at the end of which HbA1c improves from 7.8% to 7.3% and weight decreases by 3.7 kg (BMI 29.7 kg/m^2^), but nausea episodes persist.
Recommended clinical decision
Ensure that the patient is comprehensively educated on general guidelines to avoid GI AEs (Figure 1, Table 2) and maintain semaglutide at the current dose. Recommend reducing portion size and fat content of meals.
Use conditional nausea rescue therapy with oral domperidone 10 mg TID/QID upon request, and keep until the next visit if nausea persists.
Follow-up
The patient begins to apply the diet and lifestyle recommendations and starts taking domperidone before main meals. After 4 days, domperidone is not required anymore. Three months afterward, the patient reports decreased appetite without nausea. Additional decreases in HbA1c (6.8%) and weight (7% of the initial one, BMI 28.9 kg/m^2^), are achieved.
Comment
Before starting treatment with GLP-1 RA, the patient has to receive information on dietary recommendations to prevent GI AEs. If, even after applying these measures, the patient experiences nausea, temporary anti-emetic and/or prokinetic medications may be useful.
**Scenario 2. Heartburn after GLP1-RA up-titration**
Male patient, 63 y.o., BMI 34.5 kg/m^2^, with a history of gastroesophageal reflux subsequent to hiatal hernia treated with alkaline salts and with PPIs (omeprazole) for large meals. The patient comes to consultation to manage obesity. Liraglutide treatment is started following the recommended dose-escalation protocol, with weekly dose increases. When 1.8 mg OD is reached, the patient reports gastrointestinal symptoms after meals, namely postprandial heaviness, frequent eructation and heartburn, especially before bedtime.
Recommended clinical decision
Maintain liraglutide at the current dose; reinforce patient education with a focus on increasing meal frequency consisting of smaller portions; avoiding stimulant/secretory compounds (coffee, tea, spices, cola drinks) and those inducing lower esophageal sphincter relaxation (fat, fried or processed foods, tomato, mint, chocolate); avoiding alcoholic and carbonated drinks; avoiding lying down shortly after meals.
Recommend PPI use for a short time to relieve gastroesophageal reflux symptoms before continuing with dose escalation.
Escalate liraglutide at 2-week intervals between a dose level and the immediately higher one.
Follow-up
Shortly after following the advised habits and pharmacological treatment, gastroesophageal reflux symptoms notably improved. After 2 weeks of maintaining liraglutide at 1.8 mg OD, dose-escalation is resumed at 2-week intervals, until the maximum effective dose of 3 mg OD is reached. Six months after treatment start, weight loss is 8.2% (BMI 31.7 kg/m^2^).
Comment
Gastroesophageal reflux is common in obesity. GLP-1 RAs have occasionally been described to exacerbate its symptoms, possibly because of the transient delay in gastric emptying subsequent to treatment initiation [78,79]. Suitable diet habits and PPI use are usually enough to relieve symptoms. Furthermore, these improve as weight decreases. Thus, there is no need to keep pharmacological support for a long time.
**Scenario 3. Elderly patients with GI AEs subsequent to GLP-1 RA initiation**
Female patient, 80 y.o., 98 kg, BMI 39.2 kg/m^2^, T2D with HbA1c 7.2%. The current treatment is metformin 850 mg BID/sitagliptin 100 mg OD. Weight loss is required before knee prosthesis placement. Previous attempts were unsuccessful. Sitagliptin is suspended and s.c. semaglutide is started, escalating doses every 4 weeks. When 0.5 mg/wk is reached, moderate to severe GI AEs appear (persistent nausea, postprandial vomiting several times a week, pronounced hyporexia). Symptoms persist for 2 months. Three months after semaglutide start, weight loss is 14 kg; the patient reports overall weakness and sarcopenia is diagnosed according to chair stand and timed-up-and-go tests, dinamometry and bioelectrical impedance analysis.
Recommended clinical decision
Reduce semaglutide dose to 0.125 mg/wk. Four weeks later, increase the dose to 0.25 mg/wk, provided that no symptoms of gastrointestinal intolerance are documented.
Reinforce diet, increasing protein amount to 1.5 g/kg body weight. Suggest a home exercise program to improve muscle strength using elastic bands and small dumbbells.
Follow-up
Semaglutide is maintained at 0.25 mg/wk, and GI AEs are not reported. Sarcopenia tests improve in 2 months. After 6 months, patient weight is 82 kg (BMI 32.8 kg/m^2^), and HbA1c decreases to 5.9%. Knee surgery is undertaken with no complications.
Comment
Obesity is common in the elderly, with GI AE onset being earlier and more severe. The scarce experience with GLP-1 RA in >75 y.o. patients invites us to *start low and go slow*. Sarcopenia risk must be assessed before and during GLP-1 RA treatment since the sharp weight loss in the elderly may be in part at the expense of lean mass [80,81].
**Scenario 4. Insulin-treated elderly people with T2D, obesity and CKD**
Male patient, 76 y.o., BMI 32.0 kg/m^2^, poorly controlled T2D (HbA1c 8.9%), retinopathy and autonomic neuropathy, CKD with eGFR 25 mL/min/1.73 m^2^ and albumin-to-creatinine ratio 658 mg/g. The treatment consisted of linagliptin, dapagliflozin, atorvastatin and antihypertensive drugs. Baseline s.c. insulin degludec 0.2 U/kg before breakfast is started. Three months later, HbA1c is 7.5%. Linagliptin is suspended and s.c. semaglutide 0.25 mg/wk is started. The dose is increased to 0.5 mg/wk one month later. Two weeks later, the patient presents with 4–5 daily episodes of diarrhoea regardless of food composition.
Recommended clinical decision
Rule out infectious and inflammatory diarrhoea.
Encourage the patient to tightly adhere to diet/lifestyle guidelines (Figure 1, Table 2) to minimize GI AEs.
Extend duration of dose-escalation period: go back to 0.25 mg/wk for one month. Once diarrhoea disappears, return to 0.5 mg/wk and use it as maintenance dose. In the event of new diarrhoea episodes, prescribe loperamide until symptoms fade out.
Reduce insulin degludec dose gradually. Withdraw it completely 3 months later, when the insulin dose is <15 IU/day [82].
Follow-up
Four months after semaglutide dose is set at 0.5 mg/wk, metabolic and CKD parameters improve (HbA1c 6.1%, albumin-to-creatinine ratio 310 mg/g), and BMI decreases to 28.0 kg/m^2^. No new GI AEs are reported. The patient did not require insulin treatment.
Comment
Semaglutide exposure is not influenced by renal impairment [83], and thus it is a suitable choice for T2D patients with CKD and obesity. CKD and autonomic neuropathy cause gastroparesis, thus increasing GI AE risk. CKD increases the risk of GLP-1 RA-associated diarrhoea, with symptoms being more severe with albuminuria [84]. Despite GI AEs may be frequent in the first weeks, attempts to alleviate symptoms are worthwhile: withdrawal risk decreases; thus, patients benefit from cardiorrenal protection and efficient metabolic control, which may allow reducing the intensity of insulin treatment, thus minimizing hypoglycaemia risk [82].

BID, twice a day; BMI, body mass index; CKD, chronic kidney disease; eGFR, estimated glomerular filtration rate; GI AEs, gastrointestinal adverse events; GLP-1 RA, GLP-1 receptor agonist; HbA1c, glycated haemoglobin; PPI, proton pump inhibitor; QID, four times daily; s.c., subcutaneous; OD, once daily; T2D, diabetes mellitus type 2; TID, three times daily; wk, week; y.o., years old.

## Data Availability

The data are available upon request from the corresponding author.

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
