# Peer review of "Clinical Recommendations to Manage Gastrointestinal Adverse Events in Patients Treated with Glp-1 Receptor Agonists: A Multidisciplinary Expert Consensus"

_jcm, 2022, doi:10.3390/jcm12010145_

Round 1

Reviewer 1 Report

Nice summary of existing literature around gastrointestinal side effects of GLP1RA drugs, and opinion regarding treatment/prevention of these side effects.  Does not contribute novel data/information, but I think that this is a really nice resource, as I am not aware of other places where all of this information is included together.  

Author Response

Thank you very much for the review (there are no comments to answer).

Reviewer 2 Report

The perspective by Juan J. Gorgojo-Martinez et al ‘’Clinical recommendations to manage gastrointestinal adverse events in patients treated with GLP-1 receptor agonists: a multidisciplinary expert consensus’’ rigorously discussed the management of GI AEs in clinical practice associated with the use of GLP-1 RA from a multidisciplinary perspective. The authors formed a novel approach for patient education and procedures to minimize the frequency and intensity of GLP-1 RA-associated gastrointestinal AEs

Author Response

(The authors gave the same response as above.)

Reviewer 3 Report

Gorgojo-Martinez, Mezquita -Raya and collegues wrote clinical recommendation for healthcare providers in order to manage gastrointestinal side effects of GLP1 receptor agonists in clinical practice.

The paper is interesting and meaningful; and there are only few comments.

-          Line 55. “some of them are able to cross the blood brain barrier”: please add a reference;

-          Line 66. The reviewer suggests to erase the sentence “Management of GLP-1 RA is straightforward” because it is misleading;

-          Table 1. “Eat foods that contain water (soups, liquid yogurt, gelatin, and others)”: the reviewer suggests changing it in “healthy food that contain…”

-          Table 1. The reviewer suggests removing “fully baked fruit” or to specify that it could be eaten only during diarrhoea episodes.

-          From line 138 to 149 + Table 2 should be moved in the introduction paragraph or in a separate paragraph (before the section about the recommendations).

-          Paragraph 3.3 (uncommon AEs) deserves a separate chapter.

Author Response

Thank you very much for the review and comments, which have improved the manuscript notably. Please find the point-by-point responses below:

  • Line 55. “some of them are able to cross the blood brain barrier”: please add a reference.

Response: A new reference has been included, as suggested by the reviewer:

van Bloemendaal, L.; Ten Kulve, J.S.; la Fleur, S.E.; Ijzerman, R.G.; Diamant, M. Effects of glucagon-like peptide 1 on appetite and body weight: focus on the CNS. J. Endocrinol. 2014, 221, T1-16.

Ther rest of references have been properly re-numerated.

  • Line 66. The reviewer suggests to erase the sentence “Management of GLP-1 RA is straightforward” because it is misleading.

Response: The sentence has been erased, and the following one has been rephrased. It was: “…that the most frequent adverse events (AEs) associated with them are…”, and now it reads: “…that the most frequent adverse events (AEs) associated with GLP-1 RA are…”

  • Table 1. “Eat foods that contain water (soups, liquid yogurt, gelatin, and others)”: the reviewer suggests changing it in “healthy food that contain…”

Response: We have changed the sentence as suggested by the reviewer.

  • Table 1. The reviewer suggests removing “fully baked fruit” or to specify that it could be eaten only during diarrhoea episodes.

Response: “Fully baked fruit” has been removed.

  • From line 138 to 149 + Table 2 should be moved in the introduction paragraph or in a separate paragraph (before the section about the recommendations).

Response: We do agree with the reviewer that re-locating this paragraph and its corresponding table earlier in the text will improve the comprehension of the text. For this reason, we have created a new section, called “3. AEs most frequently associated with GLP-1 RAs”, just before the section focusing on recommendations. Sections and tables (and references) have been re-numerated properly.  

  • Paragraph 3.3 (uncommon AEs) deserves a separate chapter.

Response: This suggestion will also improve the distribution of contents. We have given this paragraph the category of separate section (new section: “5. Uncommon AEs”).